# RO-N3WS: Enhancing Generalization in Low-Resource ASR with Diverse Romanian Speech Benchmarks

## Abstract

We introduce RO-N3WS, a benchmark Romanian speech dataset designed to improve generalization in automatic speech recognition (ASR), particularly in low-resource and out-of-distribution (OOD) conditions. RO-N3WS comprises over 126 hours of transcribed audio collected from broadcast news, literary audiobooks, film dialogue, children's stories, and conversational podcast speech. This diversity enables robust training and fine-tuning across stylistically distinct domains. We evaluate several state-of-the-art ASR systems (Whisper, Wav2Vec 2.0) in both zero-shot and fine-tuned settings, and conduct controlled comparisons using synthetic data generated with expressive TTS models. Our results show that even limited fine-tuning on real speech from RO-N3WS yields substantial WER improvements over zero-shot baselines. We will release all models, scripts, and data splits to support reproducible research in multilingual ASR, domain adaptation, and lightweight deployment.

## 1 Introduction

Automatic Speech Recognition (ASR) has become a core component of modern machine learning pipelines, enabling transcription for media content, accessibility, and downstream language model training Kanade et al. (2015); Pelloin et al. (2022); Kuhn et al. (2025). Although speech-to-text systems have made remarkable progress, most high-performance models focus on English or other widely spoken languages, leaving many languages under-resourced. Romanian is one such example, where existing datasets such as Vox Populi Wang et al. (2021), Common Voice Ardila et al. (2020), FLEURS Conneau et al. (2023) or volunteer-collected corpora Stan et al. (2017); Georgescu et al. (2018; 2020); Ungureanu & Dascalu (2024) provide important resources but are often constrained by domain (parliamentary speech), recording style (read speech), or limited linguistic variation. While pretrained multilingual models such as Whisper Radford et al. (2023) and Wav2Vec 2.0 Baevski et al. (2020) offer strong zero-shot performance, their robustness under domain shift or across linguistically expressive speech remains underexplored in such settings.

In this paper, we introduce *RO-N3WS*, a benchmark Romanian speech dataset designed to support fine-grained evaluation and adaptation of ASR models. The dataset includes over *126 hours* of manually curated and transcribed speech from two major Romanian broadcast news sources. It is split into: (i) an *in-domain subset* (105 hours) consisting of studio and field news reports, and (ii) an *out-of-distribution (OOD) subset* (21 hours) comprising expressive and spontaneous speech from audiobooks, Romanian films, children's stories, and podcasts.

RO-N3WS captures a wide range of linguistic and acoustic features, including prosodic variation, named entities, deviations from phonemic spelling, diacritics, and domain-specific terminology. We benchmark several state-of-the-art models, including Whisper and Wav2Vec 2.0, under both zero-shot and fine-tuned regimes. Our experiments show that fine-tuning even small subsets of RO-N3WS leads to substantial gains in in-domain and OOD performance.

Our contributions are threefold: (i) We introduce RO-N3WS, a Romanian ASR benchmark that includes carefully aligned in-domain and OOD subsets, enabling systematic evaluation of domain robustness and generalization; (ii) We benchmark several strong pretrained ASR models and demonstrate that targeted fine-tuning on RO-N3WS leads to substantial performance gains in both in-domain

Table 1: Overview of multilingual and Romanian-specific speech datasets, highlighting Romanian-language coverage, speaker diversity, and recording types.

| Dataset | Total (h) | RO (h) | RO Spk. | Type |
|---------|-----------|--------|---------|------|
| Common Voice | 7,335 | 47 | 428 | Read, crowdsourced |
| VoxPopuli | 1,791 | 89 | 18 | Scripted, formal |
| FLEURS | 1,400 | 12 | 19 | Read, aligned |
| SWARA | 21 | 21 | 17 | Read, phonetically balanced |
| RODigits | 38 | 38 | 11 | Read, digits |
| RSC | 100 | 100 | 164 | Read, isolated words |
| Echo | 378 | 378 | 343 | Read, multi-domain |
| RO-N3WS (ours) | 126 | 126 | 100+ | Read, spontaneous, news + OOD |

and OOD scenarios; (iii) We compare model adaptation using natural vs. synthetic speech, showing that while expressive Text-to-Speech (TTS) can be beneficial, real recordings provide a consistently stronger supervision signal.

RO-N3WS provides a realistic and versatile testbed for research on low-resource ASR, domain adaptation, and data-centric evaluation. The full corpus and fine-tuned models will be released publicly upon acceptance.

## 2 RELATED WORK

Recent advances in multilingual ASR have been largely driven by powerful pre-trained models such as Whisper Radford et al. (2023) and Wav2Vec 2.0 Baevski et al. (2020), which have demonstrated strong zero-shot performance across many languages. However, their robustness to domain shift, stylistic variation, and low-resource adaptation remains an active area of research Radford et al. (2023). In this context, training and evaluating models on realistic, linguistically diverse corpora is essential for both understanding model limitations and improving generalization.

For Romanian, most existing corpora are either multilingual or narrowly scoped (see Table 1). Common Voice Ardila et al. (2020) provides 47 hours of user-submitted read speech but suffers from inconsistent quality. VoxPopuli Wang et al. (2021) and FLEURS Conneau et al. (2023) include Romanian subsets (89h and 12h, respectively) primarily designed for cross-lingual evaluation. However, these datasets lack expressive or conversational content, which limits their utility for studying adaptation in real-world scenarios.

Several Romanian-only corpora have been proposed, including SWARA Stan et al. (2017), RODigits Georgescu et al. (2018), and RSC Georgescu et al. (2020), but these are constrained to studio conditions, read speech, or narrow domains (e.g., digits, isolated words). The recent Echo dataset Ungureanu & Dascalu (2024) is more ambitious, comprising 378 hours across 14 domains. However, it is still scripted and collected via crowd-sourcing, and underrepresents spontaneous or emotionally varied speech.

In contrast, RO-N3WS provides a benchmark-ready dataset with in-domain and out-of-distribution splits. Its in-domain component captures real-world Romanian speech from TV broadcasts, including live interviews and varied speaking styles. The OOD subset introduces stylistic diversity through expressive speech from audiobooks, children's stories, film dialogue and conversational podcasts, which is important for evaluating fine-grained generalization in pre-trained models. This structure supports controlled adaptation experiments with Whisper, Wav2Vec 2.0, and related models.

Moreover, RO-N3WS includes linguistic phenomena underrepresented in prior corpora, such as named entities, prosodic variation, deviations from phonemic spelling, and emotionally charged delivery, which are crucial for data-centric ASR research and for developing robust ASR systems in low-resource settings. This makes it a valuable resource for studying model behavior under domain mismatch, synthetic vs. natural supervision, and targeted fine-tuning strategies. Quantitative

Table 2: Named entity density (entities per 100 tokens) across Romanian speech datasets. RO-N3WS exhibits the highest density overall, highlighting its lexical richness.

| Label | Fleurs | Common Voice | Vox Populi | Echo | RO-N3WS | RO-N3WS OOD |
|---|---|---|---|---|---|---|
| PERSON | 2.06 | 1.35 | 1.66 | 2.89 | **3.56** | 2.76 |
| LOC | **0.25** | 0.07 | 0.17 | 0.08 | 0.12 | 0.04 |
| NAT_REL_POL | 0.70 | 0.31 | **0.94** | 0.72 | 0.42 | 0.16 |
| PRODUCT | 1.24 | 0.28 | 0.26 | 1.04 | **1.44** | 0.87 |
| DATETIME | 0.93 | 0.45 | 0.49 | 0.83 | **1.46** | 0.50 |
| NUMERIC_VALUE | 0.81 | 0.45 | 0.67 | 1.23 | **1.61** | 0.46 |
| ORDINAL | **0.27** | 0.18 | 0.19 | 0.23 | 0.22 | 0.14 |
| MONEY | 0.08 | 0.02 | 0.06 | 0.09 | **0.17** | 0.03 |
| GPE | 0.75 | 0.23 | **1.31** | 0.54 | 0.90 | 0.17 |
| FACILITY | 0.62 | 0.13 | 0.13 | 0.54 | **0.89** | 0.33 |
| QUANTITY | **0.19** | 0.01 | 0.01 | 0.05 | 0.14 | 0.02 |
| ORGANIZATION | 0.26 | 0.24 | **0.54** | 0.32 | 0.36 | 0.09 |
| LANGUAGE | **0.08** | 0.01 | 0.01 | 0.02 | 0.01 | 0.02 |
| WORK_OF_ART | **0.02** | 0.01 | 0.01 | 0.01 | 0.01 | 0.01 |
| EVENT | **0.04** | 0.03 | 0.01 | 0.02 | 0.03 | 0.01 |
| PERIOD | **0.03** | 0.01 | 0.01 | 0.01 | 0.01 | 0.01 |
| **TOTAL** | 8.33 | 3.78 | 6.47 | 8.62 | **11.35** | 5.61 |

comparisons of named entity usage and prosodic expressiveness across corpora are provided in Section 3.

# 3 DATASET ANALYSIS

To highlight the linguistic and prosodic richness of RO-N3WS, we present two complementary analyses: named entity density, which reflects lexical content and topic specificity, and prosodic variation, which serves as a proxy for emotional expressiveness. We perform the analyses across five Romanian speech corpora: our dataset (RO-N3WS) and four widely used benchmarks, namely FLEURS, Common Voice, Vox Populi, and Echo.

**Named Entity Density.** We performed a detailed Named Entity Recognition (NER) analysis using the `ro_core_news_md` model from spaCy Honnibal et al. (2020), computing entity density as the number of named entities per 100 tokens. Results are summarized in Table 2. RO-N3WS exhibits the highest overall named entity density (11.35 entities per 100 tokens), surpassing Echo (8.62), FLEURS (8.33), Vox Populi (6.47), and Common Voice (3.78). More precisely, RO-N3WS shows: (i) 3.56 PERSON entities per 100 tokens (vs. 2.89 in Echo); (ii) 1.44 PRODUCT (vs. 1.24 in FLEURS) and 1.46 DATETIME mentions (vs. 0.93 in FLEURS); (iii) 1.61 NUMERIC_VALUE (compared to 1.23 in Echo) and 0.17 MONEY mentions (compared to 0.09 in Echo). These figures reflect the high concentration of factual and entity-rich language in RO-N3WS, making it especially valuable for evaluating ASR systems' ability to handle named entities, quantities, and temporal expressions. Even the RO-N3WS-OOD subset exhibits a non-trivial entity density of 5.61, higher than Common Voice (3.78) and close to Vox Populi (6.47).

**Prosodic Variation and Emotional Expressiveness.** We use the Parselmouth library Jadoul et al. (2018), a Python interface to Praat Boersma (2001), to extract prosodic features from each dataset: (1) Mean Pitch (F0): average vocal pitch, indicating the speaker's general vocal register; (2) Pitch Standard Deviation: variation in pitch, associated with expressive modulation; (3) Pitch Range: difference between the highest and lowest pitch values; higher ranges suggest greater emotional dynamics; (4) Mean Intensity: average loudness, often higher in emotionally charged utterances. These metrics serve as quantitative proxies for vocal expressiveness. Results are shown in Table 3. We analyze RO-N3WS and RO-N3WS-OOD separately, as they differ in style and composition. From Table 3 we can extract some key observations: (i) RO-N3WS-OOD exhibits the highest pitch

Table 3: Prosodic statistics across Romanian ASR datasets.

| Dataset | Mean Pitch (Hz) | Std. Pitch (Hz) | Pitch Range (Hz) | Mean Intensity (dB) |
|---------|-----------------|-----------------|------------------|---------------------|
| FLEURS | 172.01 | 60.00 | **394.19** | 54.37 |
| Common Voice | 151.45 | 37.21 | 212.08 | 49.92 |
| Vox Populi | **197.99** | 36.49 | 304.07 | 61.45 |
| Echo | 175.10 | 46.67 | 341.19 | 50.95 |
| RO-N3WS | 182.19 | 39.33 | 266.32 | **63.65** |
| RO-N3WS-OOD | 197.72 | **71.92** | 370.07 | 58.19 |

standard deviation among all datasets, indicating high expressive variability, especially relevant for spontaneous speech like film dialogue and conversational podcasts; (ii) RO-N3WS-OOD shows almost the highest mean pitch, very close to Vox Populi, reinforcing its expressive and animated nature; (iii) RO-N3WS (the in-domain component) displays the highest mean intensity, reflecting the projection and clarity typical of professional broadcast speech, such as news presenters and reporters. Compared to Echo, RO-N3WS offers complementary prosodic characteristics: (i) it displays a higher mean pitch (182.19 Hz vs. 175.10 Hz), aligning with the broadcast speech style used by reporters and presenters; (ii) while Echo exhibits greater pitch range and variation, RO-N3WS maintains high clarity and projection, as evidenced by its substantially higher mean intensity (63.65 vs. 50.95 dB).

## 4 RO-N3WS: CONSTRUCTION, ANNOTATION, AND SPLITS

**Broadcast Speech Collection and Segmentation**. We construct the RO-N3WS corpus by collecting spoken Romanian from two major TV news sources: *ProTV* and *Antena 1*. Recordings are scraped from the online archive [1] and YouTube channel [2], respectively, using custom workflows built in UiPath Studio. Each video is downloaded, converted to `.wav`, and processed through the Whisper model Radford et al. (2023). For higher temporal precision, we use `whisper-timestamped` to obtain word-level timestamps. Segments are extracted by greedily grouping consecutive complete sentences while keeping clip durations below 10 seconds. This ensures high utility for both training and human annotation. From the initial scraping pipeline, we obtained *42,225* segments from ProTV and *78,831* from Antena 1, totaling *207 hours* of raw broadcast audio.

**Data Cleaning and Quality Control**. To ensure high transcription accuracy, we discard files containing overlapping speech, music, disfluencies, foreign languages, Whisper hallucinations, or poor segmentation. Antena 1 segments in particular required the removal of repeated promotional content. Approximately *60%* of Antena 1 segments were discarded. The final cleaned dataset contains *32,319* files from ProTV (~54.5 hours) and *30,073* files from Antena 1 (~50.5 hours), resulting in *105 hours* of curated broadcast news speech.

**High-Quality Transcription and Annotation Protocol**. Each audio file is manually corrected and annotated by a pool of 15 trained annotators. Initial transcripts generated by Whisper serve as a first-pass hypothesis. Annotators apply the following guidelines: (i) Romanian diacritics are restored (ș, ț, ă, â, î); (ii) numbers are written out (e.g., *100* becomes *o sută - one hundred*); (iii) spoken abbreviations are expanded; (iv) named entities are kept as pronounced; (v) spelling errors are corrected. Each file is cross-checked by two annotators. This procedure ensures that transcripts are both orthographically correct and phonetically faithful. A few representative examples of the resulting annotations, along with corresponding Whisper outputs, are presented in the Appendix A.2.

**Data Splits and Reproducibility Strategy**. We perform a stratified split into training (85%), validation (10%), and test (5%) sets using a 20-fold strategy. All segments from the same long-form video are kept in the same fold to avoid speaker or context leakage. This results in *53,049* training segments (~89.4 hours), *6,219* validation (~10.4 hours), and *3,124* test segments (~5.3 hours).

**Evaluating Under Domain Shift: Out-of-Distribution Speech**. In addition to the broadcast news recordings, we augment our dataset with speech data that differs significantly in style, domain, and acoustic conditions, referred to as *out-of-distribution (OOD)* data. These recordings serve as

---

[1][https://stirileprotv.ro/video/

[2][https://www.youtube.com/@ObservatorTV

Table 4: Detailed statistics for the audio files contained in our RO-N3WS dataset.

| Split / Source | # Files | Total Length (s) | Min (s) | Max (s) | Avg (s) | Std Dev (s) |
|---|---|---|---|---|---|---|
| **In-domain (News)** | | | | | | |
| ProTV news | 32,319 | 196,405 | 3.00 | 11.98 | 6.08 | 1.57 |
| Observator news | 30,073 | 182,423 | 1.72 | 12.00 | 6.07 | 1.57 |
| All news | 62,392 | 378,828 | 1.72 | 12.00 | 6.07 | 1.57 |
| **Out-of-Distribution (OOD)** | | | | | | |
| Audio-books | 2,475 | 15,422 | 1.20 | 10.14 | 6.23 | 1.37 |
| Films | 4,723 | 29,814 | 3.00 | 11.94 | 6.31 | 1.32 |
| Stories | 2,821 | 17,029 | 0.66 | 21.46 | 6.04 | 2.11 |
| Podcasts | 1,723 | 12,061 | 0.52 | 29.18 | 7.00 | 2.76 |
| All OOD | 11,742 | 74,327 | 0.52 | 29.18 | 6.33 | 1.83 |

evaluation material to test the generalization capabilities of ASR models trained on in-domain (news) data. Following the same pipeline described in earlier subsections, we collect, clean, and annotate audio segments from three OOD sources: (1) *audiobooks*[3] containing expressive and narrative-style reading; (2) *famous Romanian films*[4] containing conversational speech in varied acoustic environments; (3) *children's stories*[5] containing expressive and prosodically rich narration aimed at young audiences; (4) *Romanian podcasts*[6], which provide unscripted, spontaneous conversations covering a wide range of topics and speaker styles, thereby introducing natural hesitations, turn-taking, and informal phrasing absent from more scripted sources. These subsets are transcribed and curated using the same Whisper+annotator protocol. They provide valuable test conditions for evaluating robustness to stylistic and domain variability. As shown in Table 4, the OOD evaluation sets comprise approximately *4.2 hours* of audiobook speech, *8.2 hours* of film dialogue, *4.4 hours* of children's storytelling, and *3.3 hours* of podcasts.

Additional audio duration statistics and per-source histograms for both in-domain and out-of-domain subsets are provided in Appendix A.1.

## 5 EXPERIMENTAL EVALUATION

We evaluate several state-of-the-art speech-to-text models on RO-N3WS in two key settings: *zero-shot* (no fine-tuning) and *supervised adaptation* (fine-tuning on RO-N3WS). To assess both transcription accuracy and domain robustness, we report word error rates (WER) on both in-domain (ProTV and Antena1 news) and OOD test sets (audiobooks, films, children's stories and podcasts).

In our experiments, we report separate WER scores for the ProTV and Antena1 subsets within the RO-N3WS test set. This distinction is intentional: although both sources belong to the same broadcast news genre, they represent *different linguistic and acoustic distributions*. Differences arise in presentation style, speaker identity, background conditions, and editorial content. Aggregating these into a single metric would mask these domain-specific variations. By reporting scores on each subset independently, we enable a more granular analysis of model performance and better highlight generalization differences across in-domain conditions. Moreover, this split enables controlled experiments on *in-domain generalization*: for instance, training on one source (e.g., ProTV) and testing on the other (e.g., Antena1), and vice versa. This provides a realistic scenario for assessing cross-distribution robustness, even when the domain remains the same.

---

[3]*Cortina – Ultimul caz al lui Poirot*

[4]*Buletin de București, Căsătorie cu repetiție, Toamna bobocilor, Iarna bobocilor, Primăvara bobocilor, Liceenii, Toate pânzele sus, Cu mâinile curate, Duelul, Revanșa, Un comisar acuză, Supraviețuitorul, Pistruiatul*

[5]https://www.youtube.com/@RomanianFairyTales

[6]https://www.youtube.com/@fainsisimplu,https://www.youtube.com/@unpodcastmisto,https://www.youtube.com/@IlincaVandici./podcasts

All experiments were executed using a high-performance compute environment equipped with an NVIDIA H100 GPU (80GB), Intel Xeon Platinum 8480+ CPU (112 cores), and 2TB of RAM. Full training hyper-parameters for Whisper and Wav2Vec 2.0 models are listed in Appendix A.3.

## 5.1 MODELS EVALUATED

We evaluate five speech-to-text systems in our experiments, divided into two categories: (i) open-source, fine-tunable models and (ii) commercial black-box APIs. The open-source models are evaluated both in a zero-shot setting and after supervised fine-tuning on our RO-N3WS dataset. The commercial models are accessed via API and evaluated in zero-shot mode only.

### 5.1.1 OPEN-SOURCE MODELS

**Wav2Vec 2.0 (VoxPopuli fine-tuned for Romanian)** Baevski et al. (2020) is a self-supervised model trained on raw waveforms using contrastive learning and masking, analogous to masked language modeling in NLP. In our experiments, we use the model checkpoint `facebook/wav2vec2-base-10k-voxpopuli-ft-ro`, available via Hugging Face.[7]

**Whisper** Radford et al. (2023) is a multilingual encoder-decoder Transformer trained on 680,000 hours of supervised data across various tasks, including ASR, speech translation, and voice activity detection. We evaluate several Whisper variants, like `small` and `large` and also a variant fine-tuned on the Echo dataset. *Whisper Small fine-tuned on Echo dataset* Ungureanu & Dascalu (2024) is a publicly released model trained on a combination of approximately 356 hours of prior Romanian audio data and 378 hours of the Echo dataset, totaling over 730 hours of labeled Romanian speech. The model is based on the `small` Whisper architecture and was fine-tuned using normalized transcripts with literal number expansions. This model is included as a strong Romanian-specific baseline.

### 5.1.2 COMMERCIAL MODELS (BLACK-BOX APIS)

**Vatis**[8] is a Romanian commercial ASR service known for its strong transcription accuracy in Romanian. Its models are trained on proprietary data and optimized for production use. As the model is not open-sourced, we use its public API to obtain transcriptions.

**Microsoft Transcribe**[9] is a production-level speech-to-text system integrated into Azure Cognitive Services, optimized for multilingual transcription across general domains.

**Google Chirp** Zhang et al. (2023) officially known as the *Universal Speech Model (USM)*, is a large multilingual encoder-decoder system with 2 billion parameters. It is trained on millions of hours of audio and billions of text samples, using a conformer-based encoder architecture.

## 5.2 EVALUATION METRICS

We report *Word Error Rate (WER)* as our primary evaluation metric, computed as the minimum number of insertions, deletions, and substitutions required to transform the predicted sequence into the reference, normalized by the reference length. A challenge in using WER arises from differences in text formatting induced by language models or post-processing heuristics (such as inverse text normalization). For instance, numerical phrases such as *"o mie opt sute șaizeci și trei"/ "one thousand eight hundred sixty-three"* may be rendered as *"1.863"* by some systems. Table 5 illustrates such non-semantic discrepancies. To mitigate the influence of format-specific mismatches, we construct multiple reference annotations that account for acceptable formatting variations. In this relaxed setting, we avoid penalizing semantically equivalent but syntactically different outputs.

## 5.3 ZERO-SHOT EVALUATION

In the zero-shot setting, models are evaluated without any adaptation to the RO-N3WS dataset. This setup assesses how well each system generalizes to Romanian broadcast speech and stylistically

---

[7] https://huggingface.co/facebook/wav2vec2-base-10k-voxpopuli-ft-ro
[8] https://vatis.tech
[9] https://azure.microsoft.com/en-us/products/cognitive-services/speech-to-text

Table 5: Examples where WER is high due to formatting differences, despite semantic equivalence between output and reference. Outputs from commercial ASR APIs are post-processed to remove punctuation, normalize capitalization, and standardize spacing before WER computation.

| Transcription (Romanian) | WER |
|---|---|
| **Ground-truth annotation:** de la întâi ianuarie salariul minim va fi trei mii de lei brut iar în mână un angajat ar urma să primească o mie opt sute șaizeci și trei de lei *("from the first of January, the minimum salary will be three thousand gross lei, and in hand, an employee would receive one thousand eight hundred sixty-three lei")* | 0.000 |
| **Vatis output:** De la 1 ianuarie, salariul minim va fi 3.000 lei brut, iar în mână un angajat ar urma să primească 1.863 lei. | 0.387 |
| **Microsoft Transcribe output:** De la 1 ianuarie, salariul minim va fi 3.000 RON brut, iar în mână un angajat ar urma să primească 1.863 RON. | 0.451 |

Table 6: Zero-shot Word Error Rate (WER %) on in-domain and OOD test sets. Lower is better.

| Model | In-Domain (RO-N3WS) | | Out-of-Distribution (OOD) | | | |
|---|---|---|---|---|---|---|
| | ProTV | Antena1 | Audiobooks | Films | Stories | Podcasts |
| W2V2 | 27.7 | 25.6 | 40.8 | 75.4 | 54.1 | 36.3 |
| Whisp-S | 31.6 | 26.8 | 40.0 | 60.0 | 41.1 | 31.9 |
| Whisp-L | 12.3 | 5.9 | 14.8 | 27.3 | 10.9 | 11.7 |
| Whisp-S + Echo | 9.4 | 10.1 | 18.8 | 54.1 | 21.0 | 21.6 |
| Microsoft Transcribe | 2.9 | 4.8 | 10.6 | 31.1 | 17.6 | 11.5 |
| Google Chirp (USM) | 12.1 | 11.3 | 20.2 | 37.6 | 22.4 | 22.4 |
| Vatis | 5.2 | 4.4 | 13.0 | 31.2 | 16.0 | 10.2 |

diverse OOD scenarios. Results are reported separately for in-domain (ProTV, Antena1) and OOD test sets (audiobooks, films, children's stories, podcasts) in Table 6. Overall, the results highlight substantial variation in zero-shot performance across both models and evaluation domains.

**In-domain generalization.** Among open-source models, *Whisper Large* and *Whisper Small + Echo* outperform Wav2Vec 2.0 by a large margin. Whisper Large benefits from scale (1.55B parameters), while Whisper Small + Echo demonstrates the effectiveness of Romanian-specific fine-tuning, even outperforming Whisper Large on ProTV (9.4% vs. 12.3% WER). However, the best overall in-domain performance is achieved by commercial systems: *Microsoft Transcribe* on ProTV (2.9%) and *Vatis* on Antena1 (4.4%), indicating the advantage of proprietary data and domain tuning.

**Out-of-distribution robustness.** All models degrade significantly on OOD sets, especially on *films*, where acoustic variability and overlapping dialogue present major challenges. *Whisper Large* is the most robust open-source model across OOD domains, achieving 10.9% WER on children's stories and 14.8% on audiobooks. Surprisingly, *Whisper Small + Echo* performs competitively on expressive speech (e.g., 21.0% on stories) but struggles with film audio, likely due to the mismatch between Echo's read style and the spontaneous film domain.

While multilingual models like Whisper show strong zero-shot capabilities, domain mismatch remains a significant hurdle. Romanian-specific fine-tuning (e.g., via Echo) improves in-domain performance, but robust domain generalization requires greater stylistic diversity, precisely what RO-N3WS offers.

Table 7: WER (%) of ASR models fine-tuned on RO-N3WS and its subsets. Lower is better.

| Model | In-Domain (RO-N3WS) | | Out-of-Distribution (OOD) | | | |
|---|---|---|---|---|---|---|
| | ProTV | Antena1 | Audiobooks | Films | Stories | Podcasts |
| W2V2 + RO-N3WS | 9.8 | 12.1 | 25.6 | 62.4 | 33.6 | 22.8 |
| W2V2 + ProTV | 10.5 | 17.5 | 31.8 | 69.6 | 40.5 | 30.6 |
| W2V2 + Antena1 | 14.5 | 13.4 | 28.2 | 62.3 | 36.2 | 23.8 |
| Whisp-S + RO-N3WS | 4.1 | 6.4 | 18.9 | 43.3 | 21.1 | 14.1 |
| Whisp-S + ProTV | 5.8 | 11.1 | 27.5 | 59.6 | 33.6 | 30.6 |
| Whisp-S + Antena1 | 9.6 | 7.7 | 21.3 | 46.3 | 25.8 | 16.0 |
| Whisp-S + Echo + RO-N3WS | 4.0 | 6.1 | 17.6 | 42.3 | 20.0 | 14.8 |
| Whisp-L + RO-N3WS | 2.9 | 4.4 | 13.6 | 31.7 | 14.0 | 11.1 |
| Whisp-S + ProTV (4 runs) | 6.0 ± 0.2 | 11.0 ± 0.1 | 25.2 ± 1.9 | 55.8 ± 2.9 | 30.8 ± 2.8 | 22.9 ± 1.0 |
| Whisp-S + Antena1 (4 runs) | 9.7 ± 0.1 | 7.9 ± 0.2 | 21.2 ± 0.3 | 47.8 ± 1.7 | 24.8 ± 1.7 | 16.7 ± 1.0 |

## 5.4 SUPERVISED FINE-TUNING SCENARIO

We fine-tune Whisper and Wav2Vec 2.0 models on the RO-N3WS training and validation sets. Table 7 reports WER scores across in-domain and OOD test sets. Fine-tuning improves performance over the zero-shot baseline, confirming the impact of targeted adaptation on Romanian ASR.

**Impact of RO-N3WS fine-tuning.**    Fine-tuning on RO-N3WS yields large gains across models and domains. For instance, *Whisper Small* improves from 31.6% (zero-shot ProTV) to 4.1%, and from 41.1% to 21.1% on children's stories. Wav2Vec 2.0 similarly drops from 27.7% to 9.8% on ProTV. These gains confirm the importance of in-domain supervision, even with limited data.

**Best in-domain performance.**    *Whisper Large + RO-N3WS* achieves the lowest WERs on ProTV (2.9%) and Antena1 (4.4%). However, on OOD sets, performance slightly degrades compared to the zero-shot setting (e.g., 14.0% vs. 10.9% on stories), suggesting mild overfitting to the news domain.

**Effect of source-specific fine-tuning.**    Training on only one news source hurts generalization. *Whisper Small + ProTV* performs well on ProTV (5.8%) but drops to 11.1% on Antena1. Similarly, *Whisper Small + Antena1* degrades when tested on ProTV. This confirms that although both are broadcast news, their stylistic and acoustic differences warrant separate consideration. Learning curves analyzing model performance as training data increases are shown in Appendix A.4.

**Echo pretraining improves OOD robustness.**    Pretraining on Echo yields modest yet consistent gains in OOD performance. *Whisper Small + Echo + RO-N3WS* achieves better WERs on all OOD subsets compared to Whisper Small + RO-N3WS alone (20.0% vs. 21.1% on stories), despite Echo pretraining involving 730 hours and RO-N3WS only 100 hours. This suggests RO-N3WS provides high supervision efficiency, likely due to its linguistic complexity and realistic speech delivery.

**Wav2Vec 2.0: Moderate gains.**    Wav2Vec 2.0 benefits from RO-N3WS fine-tuning but remains less robust on OOD domains (62.4% on films), reflecting its limited capacity and monolingual pretraining. Compared to Whisper models, it shows lower adaptation efficiency under distributional shift.

**Multi-run variability and error analysis.**    To assess the stability of fine-tuning results under limited-resource settings, we conducted four independent runs of *Whisper Small + ProTV* and *Whisper Small + Antena1*, each with different random seeds. The last two rows of Table 7 report the mean and standard deviation of WER across all test subsets. We observe low variability on in-domain sets (standard deviations typically below 0.2–0.3%), indicating stable fine-tuning outcomes. However, deviations are higher on OOD test sets, particularly for films and children's stories, suggesting increased sensitivity to initialization in more acoustically and stylistically diverse conditions.

Table 8: WER (%) for models fine-tuned on natural, synthetic, and mixed subsets. All use Whisper Small.

| Model | In-Domain (RO-N3WS) | | Out-of-Distribution (OOD) | | | |
|---|---|---|---|---|---|---|
| | ProTV | Antena1 | Audiobooks | Films | Stories | Podcasts |
| Zero-shot Whisp-S | 31.6 | 26.8 | 40.0 | 60.0 | 41.1 | 31.9 |
| Synthetic only (TTS) | 24.3 | 22.5 | 34.2 | 58.5 | 39.5 | 33.0 |
| Mixed (50% TTS) | 19.0 | 20.5 | 33.8 | **55.7** | 36.7 | 31.5 |
| Natural only (RO-N3WS) | **16.6** | **18.1** | **31.0** | 58.1 | **33.3** | **27.1** |

## 5.5 NATURAL VS. SYNTHETIC SUPERVISION

To evaluate whether high-quality synthetic speech can substitute or complement natural supervision, we conducted a controlled TTS fine-tuning experiment using expressive voices from ElevenLabs.[10] We synthesized audio for approximately 4 hours of transcribed text from RO-N3WS and created three training configurations using Whisper Small:

- **Natural only:** 3h real speech (train) + 1h real speech (validation);

- **Synthetic only:** 3h TTS audio + 1h TTS validation;

- **Mixed:** 1.5h real + 1.5h TTS (train), 0.5h real + 0.5h TTS (validation).

Table 8 reports WERs on the in-domain and OOD test sets of the RO-N3WS dataset. These results highlight several key findings: (1) Models fine-tuned exclusively on natural speech outperform synthetic and zero-shot baselines across all domains, underlining the value of prosodic and contextual cues in human recordings; (2) Synthetic training improves markedly over zero-shot Whisper, demonstrating that expressive TTS, when well designed, can be a viable resource in low-resource settings. However, it lags behind real data, particularly on emotionally rich speech (like children's stories); (3) A mix of natural and synthetic speech narrows the gap with natural-only models and even surpasses it on acoustically diverse domains (e.g., films). This suggests that high-quality TTS augmentation can enhance robustness, especially under domain shift.

These results suggest that while expressive synthetic speech is useful for low-resource ASR, natural recordings continue to offer unmatched supervision fidelity. Mixed pipelines offer a promising compromise, especially when scaling high-quality datasets is costly or slow. Future work could explore curriculum learning or adaptive mixing strategies to further leverage synthetic augmentation.

## 6 CONCLUSIONS AND FUTURE WORK

We introduced RO-N3WS, a high-quality Romanian ASR dataset derived from broadcast news and augmented with out-of-distribution subsets spanning audiobooks, films, children's stories and podcasts. Our benchmark demonstrates that even moderate-scale, linguistically rich supervision can substantially improve performance under both in-domain and domain-shifted conditions.

Looking ahead, RO-N3WS opens several promising directions. Expanding beyond broadcast speech to increasing speaker and dialectal diversity, and scaling OOD coverage with noisy or user-generated content would further enhance model robustness. Our preliminary results with expressive TTS and Romanian-specific pretraining also point to opportunities in synthetic augmentation, curriculum learning, and multilingual transfer.

We hope RO-N3WS will serve as a foundation for advancing robust ASR in Romanian and inspire broader work in low-resource modeling, zero-shot adaptation, and data-centric evaluation.

---

[10]https://www.elevenlabs.io

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

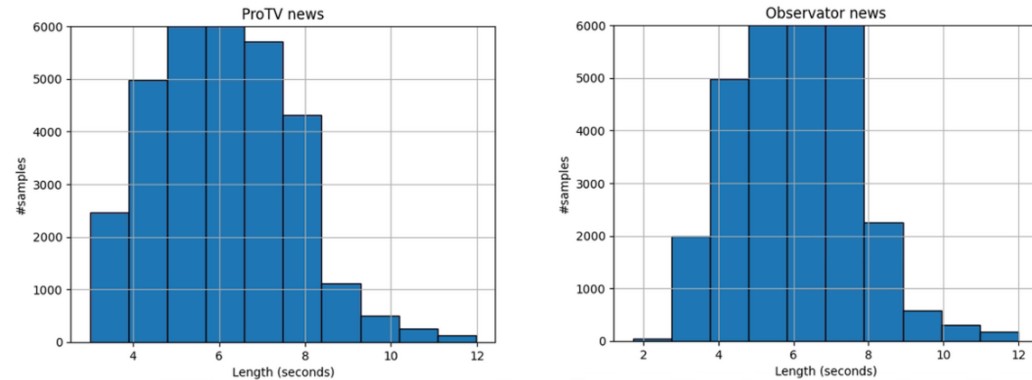

Figure 1: *Recording-duration histograms (in seconds) of collected audio files from ProTV News (left) and Observator News (right).*

# A APPENDIX

This appendix provides additional details complementing the paper. Specifically, we cover the following four topics:

1. **Dataset analysis.** We provide recording-duration histograms for both in-domain and out-of-domain splits of the RO–N3WS corpus.

2. **Annotation Examples:** We include representative samples from our manual annotation pipeline, comparing Whisper-generated transcripts with the final corrected ground-truth annotations.

3. **Training configuration.** Consolidated tables list the key hyper-parameters used to fine-tune Wav2Vec 2.0 and both Whisper variants (*Small* and *Large*).

4. **Learning curves.** We plot the word-error rate (WER) on the *ProTV News* and *Observator News* test sets as the amount of supervised data increases (5, 10, and all 17 available chunks).

## A.1 DATASET ANALYSIS

Figure 1 shows the recording-duration distribution (in seconds) for the two bulletins, *ProTV* and *Antena 1* news. Both histograms look similar, indicating that our collection and cleaning pipeline produces recordings of comparable lengths from both sources.

Figure 2 plots the recording duration distribution (in seconds) for the out-of-domain (OOD) splits: *audiobook*, *films*, *children's stories* and *podcasts*. Within the *films* subset, after grouping related titles into series, the number of recordings per series is:

- *Buletin de București*, *Căsătorie cu repetiție*: 377;
- *Toamna bobocilor*, *Iarna bobocilor*, *Primăvara bobocilor*: 442;
- *Liceenii*: 641;
- *Toate pânzele sus*: 1 297;
- *Cu mâinile curate*, *Duelul*, *Revanșa*, *Un comisar acuză*, *Supraviețuitorul*: 1239;
- *Pistruiatul*: 727.

## A.2 DATASET ANNOTATION EXAMPLES

As detailed in Section 4, our annotation process ensures that transcripts are both orthographically accurate and phonetically faithful to the original speech. Table 9 presents a few representative examples, highlighting the differences between Whisper-generated outputs and the corresponding corrected annotations.

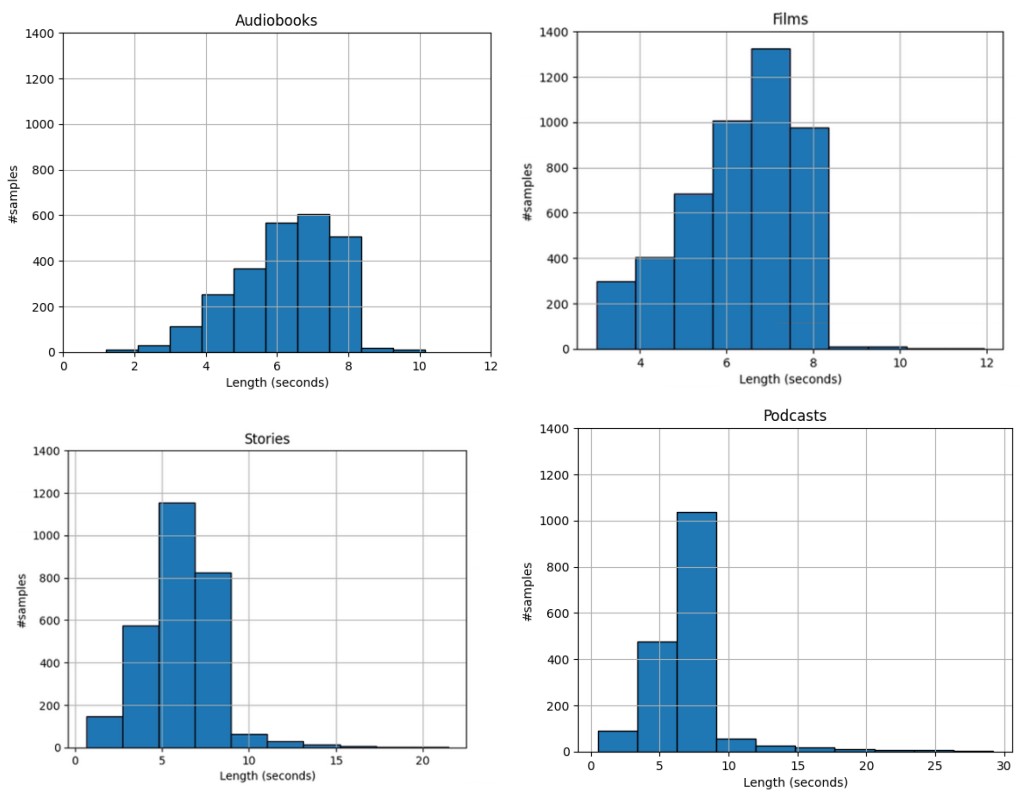

Figure 2: *Recording-duration histograms (in seconds) for out-of-distribution subsets: audiobooks, Romanian films, children's stories and podcasts.*

Table 9: Examples of Whisper-generated transcripts and their corrected annotations. The English translation corresponds to the ground-truth version.

| Whisper annotation (Romanian) | Ground-truth annotation (Romanian) | English translation (ground-truth) |
|---|---|---|
| Polițistii din Cluj au dat de urma celor care în urmă cu o lună au furat din casieria Universității de Științe Agricole 90.000 de lei | polițistii din cluj au dat de urma celor care în urmă cu o lună au furat din casieria universității de științe agricole nouăzeci de mii de lei | police in cluj tracked down those who a month ago stole ninety thousand lei from the university of agricultural sciences. |
| După ce au sequestrat și legat bine pe cei doi proprietari în vârstă, hoții șase la număr | după ce i-au sechestrat și legat bine pe cei doi proprietari în vârstă hoții șase la număr | after tying up the two elderly homeowners the six robbers... |
| Oamenii dintr-un oras din sudul Italii au avut ocazia sa asiste la filmarile pentru cea mai noua pelicula din seria James Bond | oamenii dintr-un oraș din sudul italiei au avut ocazia să asiste la filmările pentru cea mai nouă peliculă din seria james bond | people in a town in southern italy had the chance to witness the filming of the latest james bond movie. |

## A.3  TRAINING CONFIGURATION

Table 10 lists the principal hyper-parameters for the Whisper experiments reported in Tables 6, 7 and 8 in the main paper. It covers three training configurations:

- *Whisper-small* fine-tuned on a narrow subset that contains only *ProTV* or *Antena 1* news bulletins;
- *Whisper-small* fine-tuned on the full RO-N3WS corpus;
- *Whisper-large* fine-tuned on the same complete RO-N3WS dataset.

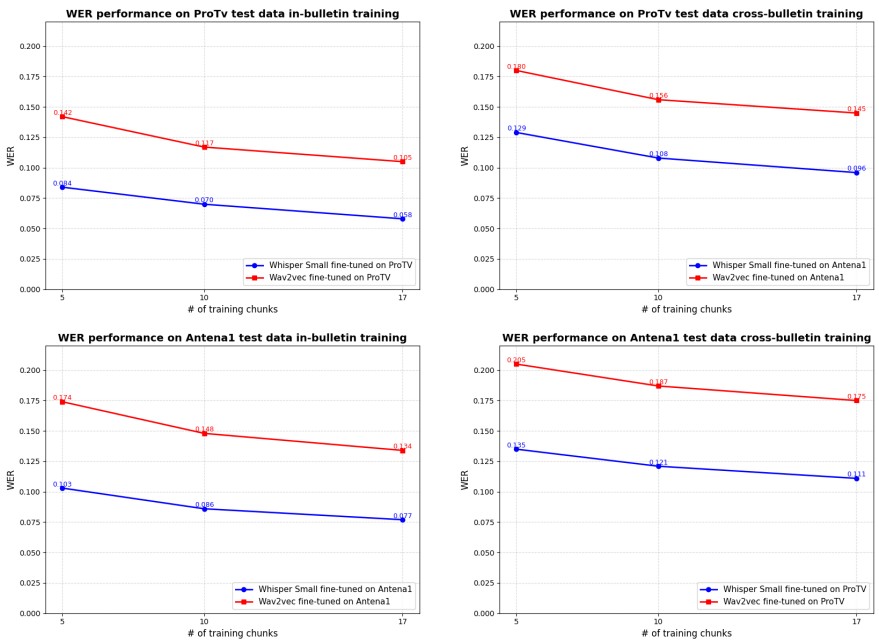

Figure 3: *Learning curves on the ProTV and Antena 1 test sets. WER is reported after fine-tuning Wav2Vec 2.0 and Whisper Small on 5, 10, and the full 17 training chunks of either the same bulletin or the other bulletin. Best viewed in color.*

Table 10: Key hyper-parameters for the Whisper configurations.

| Hyper-parameter | Whisper-small (ProTV / Antena 1) | Whisper-small (RO-N3WS) | Whisper-large (RO-N3WS) |
|---|---|---|---|
| Per-device batch size | 16 | 32 | 32 |
| Gradient-accumulation steps | 1 | 2 | 2 |
| Initial learning rate | $1 \times 10^{-5}$ | $5 \times 10^{-5}$ | $2 \times 10^{-5}$ |
| LR scheduler | Linear | Linear | Linear |
| Warm-up steps | 300 | 500 | 3 000 |
| Total training steps | 5 000 | 20 000 | 20 000 |

Table 11: Key hyper-parameters for fine-tuning *Wav2Vec 2.0*

| Hyper-parameter | Value |
|---|---|
| Per-device batch size | 16 |
| Gradient-accumulation steps | 2 |
| Initial learning rate | $5 \times 10^{-5}$ |
| Weight decay | 0.01 |
| LR scheduler | Linear (default) |
| Warm-up steps | 500 |
| Training epochs | 15 |

Table 11 details the principal hyper-parameters used to fine-tune our Wav2Vec 2.0 model on both the full RO-N3WS corpus and the narrower *ProTV* or *Antena 1* subsets.

## A.4 LEARNING CURVES

Figure 3 plots the word-error rate (WER) on the *ProTV* and *Antena 1* test sets as the amount of supervised data grows. Two systems are compared: Wav2Vec 2.0 and Whisper Small. For each

model, we draw: (i) an *in-bulletin* curve, where the system is fine-tuned on the same news bulletin it is evaluated on; (ii) a *cross-bulletin* curve, where the system is fine-tuned on the other bulletin.

Both the *ProTV* and *Antena 1* subsets of the RO-N3WS corpus are split into 17 training chunks, 2 validation chunks, and 1 test chunk. Consequently, the x-axis reports WER after fine-tuning on 5, 10, and all 17 training chunks.

Across all data regimes, Whisper Small consistently achieves lower WER than Wav2Vec 2.0. As expected, in-bulletin fine-tuning (ProTV→ProTV, Antena 1→Antena 1) outperforms the cross-bulletin setting (Antena 1→ProTV, ProTV→Antena 1), underscoring the value of a close distributional match between training and evaluation data.

