# OpenReview forum: "RO-N3WS: Enhancing Generalization in Low-Resource ASR with Diverse Romanian Speech Benchmarks"
_ICLR.cc/2026/Conference — ICLR 2026 Conference Withdrawn Submission_

### Official Review · Reviewer_s8vc · 2025-10-25

**Soundness:** 2
**Presentation:** 3
**Contribution:** 2
**Rating:** 2
**Confidence:** 4

**Summary:**

This paper introduces RO-N3WS, a new Romanian speech dataset (≈126 hours) aimed at enhancing generalization and robustness in low-resource automatic speech recognition (ASR). The dataset combines broadcast news (in-domain) and out-of-distribution (OOD) speech (audiobooks, films, children’s stories, podcasts), covering a broad stylistic and acoustic range. The authors benchmark Whisper and Wav2Vec 2.0, as well as commercial ASR systems (Vatis, Microsoft, Google Chirp), across zero-shot and fine-tuned regimes.
They show that even moderate-scale, linguistically rich supervision can substantially improve performance under both in-domain and domain-shifted conditions.

**Strengths:**

- Dataset contribution. RO-N3WS fills a clear gap in the Romanian ASR landscape, ffering diverse, well-curated data that spans broadcast, expressive, and conversational speech—missing in prior corpora.
- Benchmarks cover both open-source (Whisper, Wav2Vec 2.0) and commercial systems under both zero-shot and fine-tuned settings. The experiments are thorough.
- Good to see realistic generalization testing. The OOD sets (films, podcasts, audiobooks) provide a robust and practical way to test adaptation beyond the clean broadcast speech domain.

**Weaknesses:**

- The investigation taken in this paper is mainly empirical, using well-known techniques, with minor contributions in models and methods.
- Limited linguistic diversity. Although RO-N3WS is valuable, it is monolingual (Romanian).
- The data size of 126 hours is modest.

**Questions:**

see above.

---

### Official Review · Reviewer_i817 · 2025-10-30

**Soundness:** 2
**Presentation:** 2
**Contribution:** 1
**Rating:** 2
**Confidence:** 4

**Summary:**

This paper introduces RO-N3WS, a high-quality Romanian speech recognition (ASR) benchmark dataset comprising over 126 hours of manually curated and transcribed audio.

The paper thoroughly analyzes its method, demonstrating its linguistic richness via higher Named Entity Density and its acoustic diversity via prosodic analysis (e.g., high pitch standard deviation in the OOD subset) compared to existing Romanian ASR corpora.

The authors benchmark several ASR models in zero-shot and fine-tuned settings, showing that targeted fine-tuning on the real-world, diverse RO-N3WS data yields substantial Word Error Rate (WER) improvements, particularly enhancing robustness in OOD scenarios.

**Strengths:**

1.	The motivation behind the paper is good, targeting to improve robustness for Romanian, a relatively low-resource language.
2.	The dataset analysis provides critical insights for Romanian audio features.

**Weaknesses:**

1.	The contribution is minimal, as the work is largely confined to standard data acquisition such as web crawling, annotating, and cleaning, and is only targeted at one language.
2.	The paper does not adequately discuss or compare with other related work that is potentially heavily relevant (see Questions).
3.	The dataset size of 126 hours remains relatively small, even for a low-resource language.

**Questions:**

1.	Which version of Whisper was utilized?
2.	Why were other existing Romanian-specific models, such as RobinASR [1], not specified and compared with in this paper?
3.	If most of the dataset was gathered via web scraping/collection, why was a larger volume of Romanian data not acquired?

**References**

[1] https://arxiv.org/pdf/2111.12028

**Details Of Ethics Concerns:**

A significant concern arises regarding the use of various internet resources (models, videos, podcasts, and audiobooks) without specifying the license or copyright status of any of these materials.

This omission is particularly problematic for possibly copyrighted content, such as audiobooks and YouTube videos; therefore, an ethical review is necessary.

---

### Official Review · Reviewer_rmL4 · 2025-10-30

**Soundness:** 2
**Presentation:** 3
**Contribution:** 2
**Rating:** 4
**Confidence:** 4

**Summary:**

The authors introduce RO-N3WS, a 126-hour Romanian speech dataset designed to benchmark ASR models, to finetune existing models on it for better Romanian performance.

It is split into an in-domain (news) set and a diverse out-of-distribution (OOD) set (films, podcasts, etc.).

They use this dataset to benchmark models like Whisper, showing that while zero-shot performance is decent, fine-tuning improves in-domain scores but can hurt OOD generalization.

**Strengths:**

* New dataset for Romanian speech.
* Quantitative analysis of the dataset (e.g., NER density, prosodic features), showing it is more lexically and stylistically diverse than existing corpora
* Solid benchmark, including open-source and commercial models
* Data, models and script are to be released

**Weaknesses:**

* Critically Narrow Scope (specifically for ICLR): The entire contribution is scoped to a single, low-resource language (Romanian). It might be a better fit on a speech recognition conference.
* YODAS not mentioned / compared.
* The TTS experiments should be done differently (see comments below).
* Finetuning can probably be improved (see comments below).
* Some smaller aspects are not clear (see questions below).

**Questions:**

YODAS should also be mentioned (eg Table 1), it contains also some Romanian.

W2V2 from Table 6: This is just an unsupervised model. That cannot do speech recognition. Or do you actually train some CTC or other kind of model on top? On what data do you train that? That's misleading to put it into the zero-shot table?

I wonder a bit why you put the commercial models into table 6, but not table 7. Why to categorize them as "zero-shot". They are definitely trained on Romanian data. Maybe there is even some overlap of RO-N3WS and what they use (you don't know that). I think it's also interesting to compare those commercial models to what you have in Table 7.

Sec 5.5 / Table 8: That is not really the relevant experiment for utilizing TTS data. The much more relevant question is whether TTS can help you in addition on top of using all the RO-N3WS data. So the TTS should also not use the transcriptions, but some other text data (take some other text corpora for Romanian).

Sec 5.5  / table 8 evaluation: The validation set should always be the same, to be able to really compare the numbers. The validation on natural speech is the most relevant metric for all cases. If you want, you could also additionally provide the validation on TTS, but that's less needed.

You are finetuning a very large model (Whisper L) with very little amount of data (105h). You also mention overfitting for the OOD setting. Wouldn't it make more sense to use LoRA or so?

**Details Of Ethics Concerns:**

.

---

### Official Review · Reviewer_wewE · 2025-10-31

**Soundness:** 4
**Presentation:** 4
**Contribution:** 3
**Rating:** 8
**Confidence:** 4

**Summary:**

This paper introduces a Romanian benchmark speech dataset consisting of data from two major Romanian news outlets and data from other domains such as audio books, films and podcasts. These data have been annotated and speech to text results are presented for several models including whisper and Wav2vec 2.0. Results for out-of-distribution performance are also presented.

**Strengths:**

This is a well written paper and the dataset and code will be useful to researchers in the area of automatic speech recognition.

The exploration of the robustness to out of distributions scenarios is good to see

The significant annotation effort and attention to data quality reported will increase the utility of the dataset

**Weaknesses:**

Issues related to dataset permissions have not been addressed and copyright issues related to broadcast news should be discussed.

The motivation for focusing on broadcast news should be elaborated. What use cases are envisioned for systems trained using this dataset.

**Questions:**

How is speaker leakage prevented given the relatively small number of speakers in broadcast news data? It is doubtful that keeping data from the same long form video in the same fold is sufficient.

**Details Of Ethics Concerns:**

Permission to use the broadcast news data should be explicitly stated.

---

### Official Review · Reviewer_BGvn · 2025-11-01

**Soundness:** 3
**Presentation:** 3
**Contribution:** 2
**Rating:** 4
**Confidence:** 4

**Summary:**

The paper introduces RO‑N3WS, a 126‑hour Romanian ASR benchmark designed to probe domain robustness and low‑resource adaptation. The corpus consists of 105h in‑domain broadcast news (ProTV, Antena 1) and 21h out‑of‑distribution (OOD) material (audiobooks, film dialogue, children’s stories, podcasts). The authors provide:

(i) linguistic analyses—notably higher named‑entity density than prior Romanian corpora (e.g., FLEURS, VoxPopuli, Echo

(ii) prosodic analyses via Parselmouth/Praat, showing that the OOD split exhibits strong pitch variability, while broadcast speech shows higher intensity.

Experiments benchmark wav2vec 2.0, Whisper, a Romanian fine‑tuned Whisper‑Small (Echo), and commercial APIs (Microsoft Transcribe, Vatis, Google USM/Chirp). Metrics are WER with a “relaxed” scoring procedure to mitigate formatting mismatches (e.g., numerals vs. words). Results show sizable zero‑shot gaps across domains and large gains from fine‑tuning on RO‑N3WS (e.g., Whisper‑Small from 31.6%→4.1% WER on ProTV; Whisper‑Large + RO‑N3WS reaches 2.9%/4.4% on ProTV/Antena 1). Additional experiments compare natural vs. expressive TTS supervision and mixed pipelines, with natural-only > mixed > synthetic-only > zero‑shot, but with some robustness benefits from mixing for OOD.

**Strengths:**

The strengths of this course are:
1. Novelty: A benchmark‑ready Romanian corpus explicitly structured for in‑domain vs. OOD evaluation.
2. Solid experimental design: multiple model families (wav2vec 2.0, Whisper), model sizes, a Romanian‑specific baseline, and commercial APIs; per‑domain WER reporting (ProTV vs. Antena 1; four OOD types). Includes multi‑run variability for selected fine‑tuning setups and detailed hyperparameters. The scoring discussion (formatting mismatches, numbers) shows awareness of ASR evaluation pitfalls.
3. Clarity: Clear dataset construction and annotation protocol (diacritics restoration, numbers in words, cross‑checking).
Tables are readable and directly answer claims (e.g., zero‑shot vs. fine‑tuned, in‑domain vs. OOD).
Appendices improve reproducibility (training configs, learning curves).

**Weaknesses:**

The weaknesses of the paper are:
1. Data Rights, Licensing, and Ethics Not Fully Specified: Broadcast news and film audio are likely copyright‑protected; the paper does not clearly state the legal basis for redistribution (e.g., licenses, permissions, time‑bounded usage, “research only,” or derivative transcription rights). OOD content sourced from YouTube (films, stories, podcasts) may have rights holders and terms of service implications. In additional, the paper should clarify PII handling and consent (e.g., handling of named individuals, minors’ voices if present, takedown policy).

2. Evaluation Protocol Ambiguity: The “relaxed reference” approach (multiple references to avoid formatting penalties) is reasonable, but it departs from standard recipes; the paper should report both strict and relaxed WER, specify exact normalization (diacritics handling, punctuation rules, numerals, currency symbols), and release the scorer for reproducibility. Commercial API outputs are post‑processed (punctuation, capitalization, spacing), but it’s unclear whether identical normalization is applied to all systems, including the open‑source baselines.

3. Limited Error Analysis and Diagnostic Metrics: given the paper’s emphasis on entities and numerals, an entity‑level WER/recall, number/date accuracy, diacritics‑sensitivity, or case‑sensitive CER analysis would strengthen conclusions. Films remain very challenging (e.g., >30–60% WER for several systems post‑fine‑tune); the paper lacks a deeper acoustic/linguistic error taxonomy (overlap, reverb, background music, multi‑speaker).

4. Reproducibility Breadth: Multi‑seed variability is reported only for two settings. Extending seed sweeps or confidence intervals to the main tables would inspire more confidence. Also, reporting compute/time/energy would be welcome.

**Questions:**

As mentioned in the weakness:
1. What is the legal framework for releasing broadcast and film audio (and transcripts)? Have you obtained permissions or established a research‑only license? Will there be a takedown process for rights holders and a PII removal policy?
2. Please also release/disclose the exact scoring script (normalizers, multiple‑reference construction) and report both strict and relaxed WER. How are diacritics treated? Are currency symbols, abbreviations, and case normalized consistently across systems?
3. Given your motivation, can you report entity‑aware metrics (e.g., PER/LOC/ORG accuracy, number/date correctness)? Even small‑scale manual evaluation would substantiate the claimed entity‑rich benefit.

---

### Note · Authors · 2025-11-21

**Comment:**

We acknowledge some of the weaknesses of our work.

**Withdrawal Confirmation:**

I have read and agree with the venue's withdrawal policy on behalf of myself and my co-authors.